# Robust Reinforcement Learning with Wasserstein Constraint

## Abstract

Robust Reinforcement Learning aims to find the optimal policy with some extent of robustness to environmental dynamics. Existing learning algorithms usually enable the robustness though disturbing the current state or simulated environmental parameters in a heuristic way, which lack quantified robustness to the system dynamics (i.e. transition probability). To overcome this issue, we leverage Wasserstein distance to measure the disturbance to the reference transition kernel. With Wasserstein distance, we are able to connect transition kernel disturbance to the state disturbance, i.e. reduce an infinite-dimensional optimization problem to a finite-dimensional risk-aware problem. Through the derived risk-aware optimal Bellman equation, we show the existence of optimal robust policies, provide a sensitivity analysis for the perturbations, and then design a novel robust learning algorithm—**W**asserstein **R**obust **A**dvantage **A**ctor-**C**ritic algorithm (WRAAC). The effectiveness of the proposed algorithm is verified in the Cart-Pole environment.

## 1 Introduction

Robustness to environmental dynamics is an important topic in safe Reinforcement Learning. Take autonomous vehicle as an example. Autonomous vehicles have to adapt the complex real-world situations, but usually it is unlikely to cover all scenarios during training in real-world environments. To handle this issue, typically, a simulated environment are employed to help build a driving agent, however, the gap between the training and target environments makes the strategies trained with simulated environments sub-optimal to the real-world scenarios (Mannor et al., 2004; 2007). Learning robust policies from simulated environments is a challenging problem for safe Reinforcement Learning.

For robust Reinforcement Learning algorithms, existing methods lie on two branches: One type of methods, borrowed from game theory, introduces an extra agent to disturb the simulated environmental parameters during training (Atkeson & Morimoto, 2003; Morimoto & Doya, 2005; Pinto et al., 2017; Rajeswaran et al., 2016). This method has to rely on the environmental characterization. The other type of methods disturbs the current state through Adversarial Examples (Huang et al., 2017; Kos & Song, 2017; Lin et al., 2017; Mandlekar et al., 2017; Pattanaik et al., 2018), which is more heuristic. Unfortunately, both methods are lack of theoretical guarantee to the robustness extent of transition dynamics.

To address these issues, we design a Wasserstern constraint, which restricts the admissible transition probabilities within a Wasserstein ball centered at some reference transition dynamics. By applying the strong duality of Wasserstein distance (Santambrogio, 2015; Blanchet & Murthy, 2019), we are able to connect the disturbance on transition dynamics with the disturbance on the current state. As a result, the original infinite-dimensional robust optimal problem is reduced to some finite-dimensional ordinary risk-aware RL problem. Through the moderated optimal Bellman equation, we prove the existence of robust optimal policies, provide the theoretical analyse on the performance of optimal policies, and design a corresponding —**W**asserstein **R**obust **A**dvantage **A**ctor-**C**ritic algorithm (WRAAC), which does not depend on the environmental characterization. In the experiments, we verified the robustness and effectiveness of the proposed algorithms in the Cart-Pole environment.

The remainder of this paper is organized as follows. In Section 2, we briefly introduce some related work in Markov Decision Processes. In Section 3, we mainly describe the framework of Wasserstein robust Reinforcement Learning. In Section 4, we propose robust Advantage Actor-Critic algorithms according to the moderated robust Bellman equation. In Section 5, we perform experiments on the Cart-Pole environment to verify the effectiveness of our method. Finally, Section 6 concludes our study and provide possible future works.

## 2 RELATED WORK

In this section, we introduce some related work in the fields of MDPs. In robust MDP, the set of all possible transition kernels is called uncertainty set, which can be defined in various ways: one choice could be likelihood regions or entropy bounds of the environment parameters (White III & Eldeib, 1994; Nilim & El Ghaoui, 2005; Iyengar, 2005; Wiesemann et al., 2013); another choice is to constrain the deviation from a reference environment through some statistical distance. For example, Osogami (2012) discussed such robust problem where the uncertainty set are defined via Kullback-Leibler divergence, and also uncover the relations between robust MDPs using $f$-divergence constraint and risk-aware MDPs.

Indeed, it was observed that since the robust MDP framework ignores probabilistic information of the uncertainty set, it can provide conservative solutions (Delage & Mannor, 2010; Xu & Mannor, 2007). Some papers consider bringing prior knowledge of dynamics to robust MDPs, and name such problem distributionally robust MDPs. Xu & Mannor (2010) discuss robust MDPs with prior information to estimate the confidence region of parameters abound, which is a moment-based constraint, and they also show that such distributionally robust problems can be reduced to standard robust MDP problems. Yang (2017; 2018) use Wasserstein distance to evaluate the difference among the prior distributions of transition probabilities. However, Yang's algorithms are not appropriate for complex situations, because they need to estimate enough transition kernels to approximate prior distribution at each step.

## 3 WASSERSTEIN ROBUST REINFORCEMENT LEARNING

In this section, we specify the problem of interest, which is actually a minimax problem constrained by some Wassserstein-based uncertainty set. We start with introducing a general theoretical framework, i.e., robust Markov Decision Process, and then briefly recall the definition of Wasserstein distance between probability measures. Inspired by the strong duality brought by Wasserstein-based uncertainty set, the robust MDP is reformulated to some risk-aware MDP, making connections clear between robustness to dynamics and robustness to states.

### 3.1 ROBUST MARKOV DECISION PROCESS

Unlike ordinary Markov Decision Processes (MDPs), in robust MDP, environmental dynamics, including transition probabilities and rewards, might change over time (Nilim & El Ghaoui, 2004; 2005). Theoretically, such dynamics can be treated as stochastic changes within an uncertainty set. The objective of robust MDP is to find the optimal policy under the worst dynamics.

Given discrete-time robust MDPs with continuous state and action spaces, without loss of generalization, we only consider the robustness to transition probabilities. Basic elements of robust MDPs include $(\mathcal{X}, \mathcal{A}, \mathcal{Q}, c)$, where

- $\mathcal{X}$: state space, which is a Borel measurable metric space.
- $\mathcal{A}$: action space, which is a Borel measurable space. Let $A(x) \in \mathcal{A}$ represent all the admissible actions at state $x \in X$, and $\mathbb{K}_A$ denote all the possible state-action pairs, i.e., $\mathbb{K}_A = \{(x, a) : x \in \mathcal{X}, \ a \in A(x)\}$.
- $\mathcal{Q}$: the uncertainty set that contains all possible transition kernels.
- $c$: $\mathbb{K}_A \to \mathbb{R}$, the immediate cost function. Generally we assume it is continuous and $c \in [0, \bar{c}]$ for some non-negative constant $\bar{c}$.

The robust system evolves in the following way. Let $n \in \mathbb{N}$ denote the current time and $x_n \in \mathcal{X}$ the current state. Agent chooses an action $a_n \in A(x_n)$ and environment selects a transition kernel $q_n$ from the uncertainty set $\mathcal{Q}$, respectively. Then at the next time $n + 1$, an agent observes an immediate cost $c(x_n, a_n)$ and a new state $x_{n+1} \in \mathcal{X}$ which follows the distribution $q_n(\cdot|x_n, a_n)$. The process repeats at each stage and produces trajectories in a form of $\omega = (x_0, a_0, q_0, c_0, x_1, a_1, q_1, c_1, ...)$. Let $\Omega = (\mathcal{X} \times \mathcal{A} \times \mathcal{Q} \times [0, \bar{c}])^\infty$ denote all the trajectories. Let $\Omega_n = \{\omega_n = (x_0, a_0, q_0, c_0, x_1, a_1, q_1, c_1, ..., x_n)\}$ denote all trajectories up to time $n$ and $\tilde{\Omega}_n = \{\tilde{\omega}_n = (x_0, a_0, q_0, c_0, x_1, a_1, q_1, c_1, ..., x_n, a_n)\}$ denote all trajectories up to time $n$ with action $a_n$.

Correspondingly, a randomized policy is a series of stochastic kernels: $\pi = (\pi_0, \pi_1, \pi_2, ...)$ where $\pi_n(\cdot|\omega_n)$ is a probability measure over $A(x_n)$. We name $\pi$ **primal policy** and use $\Pi$ to represent all such randomized policies. If $\pi_n(\cdot|\omega_n) = \pi_n(\cdot|x_n)$ for $n \geq 0$, we say the policy is Markov. If $\pi_n \equiv \pi_0$ for any $n \geq 0$, this policy is stationary. If there exists measurable functions $f_n : \Omega_n \to \mathcal{A}$ such that $\pi_n(f_n(\omega_n)|\omega_n) \equiv 1$, $n \geq 0$, this policy is called deterministic. We denote the set of all such deterministic, stationary, Markov policies by $\mathbb{F}$.

The selection of transition kernels can be treated as a deterministic policy deployed by a secondary adversarial agent. Let $g = (g_0, g_1, g_2, ...)$ with $g_n : \tilde{\Omega}_n \to \mathcal{Q}$ denote the **adversarial policy**. We use $\mathbb{G}$ to represent all such deterministic policies. Similarly, if $g_n(\cdot|\tilde{\omega}_n) = g_n(\cdot|x_n, a_n)$ for all $n \geq 0$, the policy is Markov, and if $g_n \equiv g_0$ for any $n \geq 0$, the policy is stationary.

Given the initial state $X_0 = x \in \mathcal{X}$, primal policy $\pi \in \Pi$ and adversarial policy $g \in \mathbb{G}$, applying the Ionescu-Tulcea theorem (Hernández-Lerma & Lasserre, 2012a; Bertsekas & Shreve, 2004), there exist a probability measure $\mathbb{P}_x^{\pi,g}$ on trajectory space. Let $\mathbb{E}_x^{\pi,g}$ denote the corresponding expectation operation.

As for the performance criterion, we consider the infinite-horizon discounted cost. Let $\gamma \in (0, 1)$ be the discounting factor. The discounted cost contributed by trajectory $\omega \in \Omega$ is $C_\gamma(\omega) = \Sigma_{n=0}^\infty \gamma^n c(x_n, a_n)$. Given the initial state $x_0 = x$, policies $\pi$ and $g$, the expected infinite-horizon discounted cost is

$$C_\gamma^{\pi,g}(x) := \mathbb{E}_x^{\pi,g}[\Sigma_{n=0}^\infty \gamma^n c(x_n, a_n)]. \tag{1}$$

Robust MDPs aim to find the optimal policy $\pi^*$ for the agent under the worst realization of $g \in \mathbb{G}$, which means that $\pi^*$ reaches

$$\inf_\pi \sup_g C_\gamma^{\pi,g}(x). \tag{2}$$

This minimax problem can be seen as a zero-sum game of two agents.

## 3.2 WASSERSTEIN DISTANCE

The popular Wasserstein distance is a special case of optimal transport costs, which measures the discrepancy between two probabilities in terms of minimum total costs associated with some transport function. For any two probability measures $Q$ and $P$ over the measurable space $(\mathcal{X}, \mathcal{B}(\mathcal{X}))$, let $\Xi(Q, P)$ denote the set of all joint distributions on $\mathcal{X} \times \mathcal{X}$ with $Q$ and $P$ are respective marginals. Each element in $\Xi(Q, P)$ is called a coupling between $Q$ and $P$. Let $\kappa : \mathcal{X} \times \mathcal{X} \to [0, \infty)$ be the transport cost function between two positions, which is non-negative, lower semi-continuous and satisfy $\kappa(z, y) = 0$ if and only if $z = y$. Intuitively, the quantity $\kappa(z, y)$ specifies the cost of transporting unit mass from $z$ in $\mathcal{X}$ to another element $y$ of $\mathcal{X}$. Then the optimal transport total cost associated with $\kappa$ is defined as follows:

$$D_\kappa(Q, P) := \inf_{\xi \in \Xi(Q,P)} \left\{ \int_{\mathcal{X} \times \mathcal{X}} \kappa(z, y) d\xi(z, y) \right\}.$$

Therefore, the optimal transport cost $D_\kappa(Q, P)$ corresponds to the lowest transport cost that can be obtained among all couplings between $Q$ and $P$. Let the transport cost function $\kappa$ be some distance metric $d$ on $\mathcal{X}$, and then it is actually the Wasserstein distance of first order. Wasserstein distance of order $p$ is defined as:

$$W_p(Q, P) := \inf_{\xi \in \Xi(Q,P)} \left\{ \int_{\mathcal{X} \times \mathcal{X}} d(z, y)^p d\xi(z, y) \right\}^{\frac{1}{p}}, \ p \geq 1.$$

Unlike Kullback-Liebler divergence or other likelihood-based divergence measures, Wasserstein distance is a proper metric on the space of probabilities. More importantly, Wasserstein distance does not restrict probabilities to share the same support (Villani, 2008; Santambrogio, 2015). Let $d(z, y) = \| z - y \|_2$, $\kappa(z, y) = \frac{1}{p} \| z - y \|_2^p$ and $\delta = \frac{1}{p}\epsilon^p$, the $\epsilon$-Wasserstein ball of order $p$ and the $\delta$-optimal-transport ball are identical:

$$\{Q : W_p(Q, P) \leq \epsilon\} = \{Q : D_\kappa(Q, P) \leq \delta\}.$$

Due to its superior statistical properties, Wasserstein-based uncertainty set has recently received a great deal of attention in DRSO problem (Gao & Kleywegt, 2016; Esfahani & Kuhn, 2018; Blanchet & Murthy, 2019), adversarial example (Sinha et al., 2017), and so on. We will apply it to robust RL.

### 3.3 MAIN RESULT

Let the uncertainty set $\mathcal{Q}$ be a $\epsilon$-Wasserstein ball of order $p$ centered at some reference transition kernel $P$:

$$\mathcal{Q} = \{Q : W_p(Q(\cdot|x, a), P(\cdot|x, a)) \leq \epsilon, \ \forall (x, a) \in \mathbb{K}_A\} \tag{3}$$
$$= \{Q : D_\kappa(Q(\cdot|x, a), P(\cdot|x, a)) \leq \delta, \ \forall (x, a) \in \mathbb{K}_A\}, \tag{4}$$

The radius $\epsilon$ or $\delta$ reflects the extent of adversarial perturbation to the reference transition kernel $P$. The difference between our theoretical framework and Yang (2017; 2018) is that our framework is trying to find the optimal solution for the worst transition kernel within the Wasserstein ball, while theirs is trying to find the optimal solution for the worst distribution over transition kernels.

Recall the state value function (1) at state $x_0$ given primal policy $\pi$ and adversarial policy $g$, we can rewrite the state value function as follows,

$$
\begin{aligned}
C_\gamma^{\pi,g}(x_0) &= \mathbb{E}_{x_0}^{\pi,g}[\Sigma_{n=0}^\infty \gamma^n c(x_n, a_n)] \\
&= \mathbb{E}_{x_0}^{a_0 \sim \pi, q_0}[c(x_0, a_0) + \mathbb{E}_{x_1 \sim q_0(\cdot|x_0, a_0)}^{(1)\pi, (1)g}[\Sigma_{n=1}^\infty \gamma^n c(x_n, a_n)]] \\
&= \mathbb{E}_{x_0}^{a_0 \sim \pi, q_0}[c(x_0, a_0) + \gamma \int_{x_1 \in \mathcal{X}} q_0(dx_1|x_0, a_0) C_\gamma^{(1)\pi, (1)g}(x_1)],
\end{aligned}
$$

where $^{(1)}\pi = (\pi_1, \pi_2, ...)$ and $^{(1)}g = (g_1, g_2, ...)$ are the shift policies. Since $c$ is continuous and bounded, the value function is actually continuous in $\mathcal{X}$ and belongs to $[0, \frac{\bar{c}}{1-\gamma}]$.

Let $u : \mathcal{X} \to \mathbb{R}$ be a measurable, upper semi-continuous function with $u \in [0, \frac{\bar{c}}{1-\gamma}]$, and let $\mathbb{U}$ denote the set of all such functions. For state $x \in \mathcal{X}$ and action $a \in A(x)$. Consider the following operator $H^a$ defined on $\mathbb{U}$:

$$(H^a u)(x) := c(x, a) + \sup_{Q \in \mathcal{Q}} \gamma \int_{y \in \mathcal{X}} Q(dy|x, a) u(y). \tag{5}$$

Applying Lagrangian method and the strong duality property brought by Wasserstein distance (Blanchet & Murthy, 2019), we reformulate (5) to the following form:

$$(H^a u)(x) = \inf_{\lambda \geq 0} c(x, a) + \gamma \lambda \delta + \gamma \int_{y \in \mathcal{X}} P(dy|x, a)[\sup_{z \in \mathcal{X}}(u(z) - \lambda \kappa(z, y))]. \tag{6}$$

The significance of this strong dual representation lies on the fact that the operator $\sup_{\mathcal{Q}}$ in eq. (5) is replaced by $\sup_{z \in \mathcal{X}}$ in eq. (6), which leads a much easier optimization algorithm. The right-hand side of eq. (6) is a normal iterated-risk function. That is, it reduces the infinite-dimensional probability-searching problem (5) into an ordinary finite-dimensional optimization problem (6).

It is easy to verify that $H^a$ maps $\mathbb{U}$ to $\mathbb{U}$. Thus, given a state $x \in \mathcal{X}$ and agent policy $\pi$, we have the following expected Bellman-form operator:

$$
\begin{aligned}
(H^\pi u)(x) &:= \int_{a \in A(x)} \pi(da|x) H^a u(x) \\
&= \inf_{\lambda \geq 0} \gamma \lambda \delta + \int_{a \in A(x)} \pi(da|x)[c(x, a) + \gamma \int_{\mathcal{X}} P(dy|x, a)[\sup_{z \in \mathcal{X}}(u(z) - \lambda \kappa(z, y))]].
\end{aligned}
$$

Similarly, $H^\pi$ maps $\mathbb{U}$ to $\mathbb{U}$ as well. Under the following Assumption 1, we are able to define the optimal iteration operator and show its contraction property.

**Assumption 1.** *$\mathcal{X}$ is a compact metric space. For any $x \in \mathcal{X}$, $A(x)$ is compact and $H^a$ is lower semi-continuous on $a \in A(x)$.*

Then, given an initial state $x \in \mathcal{X}$, the following optimal operator over $\mathbb{U}$ is well-defined.

$$(Hu)(x) := \inf_{a \in A(x)} H^a u(x) \tag{7}$$

$$= \inf_{a \in A(x), \lambda \geq 0} c(x, a) + \gamma\lambda\delta + \gamma \int_{\mathcal{X}} P(dy|x, a)[\sup_{z \in \mathcal{X}}(u(z) - \lambda\kappa(z, y))]. \tag{8}$$

It is simple to verify that $H$ maps $\mathbb{U}$ to $\mathbb{U}$. The contraction property of $H$ is shown in Lemma 1. We put the proof in the appendix.

**Lemma 1.** *$H$ is a contraction operator in $\mathbb{U}$ under $L_\infty$ norm. There exists an unique element in $\mathbb{U}$, denoted as $u^*$, satisfying $Hu^* = u^*$.*

For any $u_0 \in \mathbb{U}$, $u_n := Hu_{n-1} = H^n u_0$. Due to the contraction, we have

$$\lim_{n \to \infty} u_n = \lim_{n \to \infty} H^n u_0 = u^*, \tag{9}$$

which indicates an iterative procedure of finding the optimal value function. Based on this optimal value function, we can demonstrate the existence of optimal policies, and single out an optimal policy who is deterministic, Markov and stationary, as shown in Theorem 1. We put the proof in the appendix.

**Theorem 1.** *There exists a deterministic Markov stationary policy $f \in \mathbb{F}$ that satisfies*

$$H^f u^* = Hu^* = u^*.$$

We now obtain the existence of an unique robust optimal value function, as well as a robust optimal policies, which is deterministic, Markov and stationary. Through an iterative procedure as (9), we can design corresponding algorithms for robust Reinforcement Learning.

### 3.4 SENSITIVITY ANALYSIS

Before going to the algorithm design, we present a sensitivity analysis for the optimal value function w.r.t. the radius $\delta$ and the Wasserstein order $p$. Let $\lambda^*$ and $z^*(y, \lambda^*) = \arg\max_{z \in \mathcal{X}}(u(z) - \lambda^*\kappa(z, y))$ be a solution of equation (8), and $\lambda^*$ is non-negative.

If $\lambda^* = 0$, which means the worst transition kernel is within our fixed $\epsilon$-Wasserstein ball, equation (8) can be reduced to an ordinary problem:

$$(Hu)(x) = \inf_{a \in A(x)} c(x, a) + \gamma \sup_{z \in \mathcal{X}} u(z).$$

Thus $u^*$ has nothing to do with $\delta$ or $p$.

If $\lambda^* > 0$, via the envelop theorem, the gradient of optimal value function w.r.t. $\delta$ can be calculated as follows.

$$\frac{\partial u^*(x)}{\partial \delta} = \gamma\lambda^* > 0. \tag{10}$$

This gradient remains positive. That is, the optimal value function increases as the volume of Wasserstein ball increases (remember that $\delta = \frac{1}{p}\epsilon^p$ and the value function represents the discounted cost). Similarly, via the envelop theorem, the gradient w.r.t. $p$ can be calculated as follows.

$$\frac{\partial u^*(x)}{\partial p} = -\gamma\lambda^* \int_{\mathcal{X}} P(dy|x, a)(\log \| z^*(y, \lambda^*) - y \|_2 - \frac{1}{p})\frac{\| z^*(y, \lambda^*) - y \|_2^p}{p}. \tag{11}$$

Since $\lambda^* > 0$, the worst transition kernel $Q^*$ satisfies $W_p(Q^*, P) = \epsilon$, i.e. $D_\kappa(Q^*, P) = \delta$.[1] Notice that calculating $z^*(y, \lambda^*)$ for $y$ is actually trying to find an optimal transport map $T_p : \mathcal{X} \to \mathcal{X}$,

---

[1]Derived from the fact that if $W_p(Q^*, P) < \epsilon$, there must be $\lambda^* = 0$.

which substantially perturbs $P$ to $Q^*$. Recall that $u^*$ is upper semi-continuous and its domain is compact, and then we can actually regard $u^*$ as the Kantorovich potential (Villani, 2008) for a transport cost function $\lambda^* \kappa$ in the transport from $P$ to $Q^*$. For $p > 1$, $\lambda^* \kappa$ is strictly convex. Through theorem 1.17 in Santambrogio (2015), we can write the optimal transport map in an explicit way, as well as the gradient over $p$.

$$z^*(y, \lambda^*) = T_p(y) = y - (\lambda^*)^{-\frac{1}{p-1}} \parallel \nabla_y u^*(y) \parallel^{-\frac{p-2}{p-1}} \nabla_y u^*(y), p > 1.$$

$$\frac{\partial u^*(x)}{\partial p} = -\frac{\gamma \lambda^*}{p(p-1)} \int_{\mathcal{X}} P(dy|x, a)(\log \parallel \frac{\nabla_y u^*(y)}{\lambda^*} \parallel_2 - \frac{p-1}{p}) \cdot \parallel \frac{\nabla_y u^*(y)}{\lambda^*} \parallel_2^{\frac{p}{p-1}}, p > 1.$$

Thus when $\frac{1}{p} \leq 1 - \log \parallel \frac{\nabla_y u^*(y)}{\lambda^*} \parallel_2$ for all $y \in \mathcal{X}$, the gradient over $p$ is non-negative. Larger $\lambda^*$ makes non-negativity more likely to happen. Remember that $\lambda^*$ actually reflect the extent of robustness, i.e., larger $\lambda^*$ coincides with smaller radius $\epsilon$. Intuitively, when the volume of Wasserstein ball is very small, the extent of perturbation at each point is small with high probability, making the gradient (11) positive. Thus in such situation, smaller $p$ is preferred.

## 4    WASSERSTEIN ROBUST ADVANTANGE ACTOR-CRITIC ALGORITHMS

In reinforcement learning, the agent does not know the precise environment dynamics, i.e., the transition kernel and immediate cost function are unknown. Some researchers leverage an adversarial agent to inject perturbations into environmental parameters during training procedures (Pinto et al., 2017). However, such methods have to work with pre-defined environmental parameters, and are lack of quantified robustness toward transition kernels. Other researchers borrow the idea of adversarial examples and disturb observed states in a heuristic way (Nguyen et al., 2015). They also lose the explanation of robustness towards system dynamics.

Following the analysis in Section 3, we develop a robust Advantage Actor-Critic algorithm: a critic neural network with parameters $w$, denoted by $u_w$, is employed to estimate value function; and an actor neural network with parameters $\theta$, denoted by $\pi_\theta$, is designed as the primal policy. Rewrite equation (8):

$$(Hu_w)(x) = \inf_{\theta, \lambda \geq 0} \int_{a \in A(x)} \pi_\theta(da|x)[c(x, a) + \gamma \lambda \delta + \gamma \int_{\mathcal{X}} P(dy|x, a)[\sup_{z \in \mathcal{X}}(u_w(z) - \lambda \kappa(z, y))]].$$

**Update for $z$ and $\lambda$:** Let $f_w(z; y, \lambda) := u_w(z) - \lambda \kappa(z, y)$ where $\kappa(z, y) = \frac{1}{p} \parallel z - y \parallel^p, p \geq 1$. Given $y \in \mathcal{X}$ and $\lambda \in [0, \infty)$, denote

$$z_{y, \lambda} := \arg\max_{z \in \mathcal{X}} f_w(z; y, \lambda).$$

Initially, $z_{y, \lambda}$ can be treated as the maximum perturbation to state $y \in \mathcal{X}$, given the penalty $\lambda$. The gradient of $f_w$ over $z$ is: $\nabla_z f_w = \nabla_z u_w(z) - \lambda ||z - y||^{p-2}(z - y)$. Let $G_w(\lambda; x, a) := \lambda \delta + \int_{\mathcal{X}} P(dy|x, a)[\sup_{z \in \mathcal{X}}(u_w(z) - \lambda \kappa(z, y))]$, and we get

$$\lambda_{x, a} := \arg\min_{\lambda} G_w(\lambda; x, a).$$

Combining the envelope theorem, we can obtain the gradient of $G_w$ w.r.t. $\lambda$: $\nabla_\lambda G_w = \delta - \int_{\mathcal{X}} P(dy|x, a) \kappa(z_{y, \lambda}, y)$. The expectation in the gradient can be approximated by Monte Carlo: take action $a$ at state $x$ for $n$ times; under the reference transition kernel $P$, observe the next states $y^j$ and quadruples $(x, a, c, y^j)$, $j = 1, 2, \cdots, n$; and then we can approximate $\nabla_\lambda G_w \approx \delta - \frac{1}{n} \sum_{j=1}^{n} \kappa(z_{y^j, \lambda}, y^j)$.

**Critic Update Rule:** Given state $x \in \mathcal{X}$ and policy $\pi_\theta$, let

$$J(\theta, w, x) := \int_{a \in A(x)} \pi_\theta(da|x)[c(x, a) + \gamma G_w(\lambda_{x, a}; x, a)].$$

To calculate $J$, similarly, we leverage Monte Carlo, take actions $a_i \sim \pi_\theta(\cdot|x)$, $i = 1, 2, \cdots, m$ at the same state $x$ for $m$ times, observe $m$ "state-action" pairs $(x, a_i)$, $i = 1, 2, \cdots, m$, and then approximate $J(\theta, w, x) \approx \frac{1}{m} \sum_{i=1}^{m} [c(x, a_i) + \gamma G_w(\lambda_{x, a}; x, a_i)]$.

Let $e(x, a_i) := c(x, a_i) + \gamma G_w(\lambda_{x,a}; x, a_i) - u_w(x)$, and $e(x)$ denote the difference between the observed cost and the critic network:

$$e(x) := J(\theta, w, x) - u_w(x) \approx \frac{1}{m} \sum_{i=1}^{m} e(x, a_i) = \frac{1}{m} \sum_{i=1}^{m} [c(x, a_i) + \gamma G_w(\lambda_{x,a}; x, a_i) - u_w(x)].$$

Through the envelope theorem, we can obtain the following gradient of $e(x)$ w.r.t. $w$:

$$\nabla_w e(x) = \frac{1}{m} \sum_{i=1}^{m} \gamma \int_{\mathcal{X}} P(dy|x, a_i) \nabla_w u_w(z_{y, \lambda_{x,a}}) - \nabla_w u_w(x) \tag{12}$$

$$\approx \frac{\gamma}{mn} \sum_{i=1}^{m} \sum_{j=1}^{n} \nabla_w u_w(z_{y_i^j, \lambda_{x,a_i}}) - \nabla_w u_w(x). \tag{13}$$

Notice that we should actually update the critic network via minimizing $\frac{1}{2}e(x)^2$, and the gradient is

$$\nabla_w \frac{1}{2} e(x)^2 = e(x) \cdot \nabla_w e(x).$$

In practice, we usually can let $m = n = 1$ to obtain faster convergence.

**Actor Update Rule:** In classical AC algorithms, directly minimizing "state-action" value function $J(\theta, w, x)$ may cause large variance and slow convergence, and optimizing the advantage function is a better choice instead. The advantage function is

$$A(x, a) := c(x, a) + \gamma G_w(\lambda_{x,a}; x, a) - u_w(x) = e(x, a).$$

Thus we can find the optimal $\theta$ via minimizing the expected advantage function $A(x, \theta) = \int_{a \in A(x)} \pi_\theta(da|x) e(x, a)$. Similarly, we can approximate the gradient of $A$ w.r.t. $\theta$ as follows:

$$\nabla_\theta A(x, \theta) = \int_{a \in A(x)} \pi_\theta(da|x) \nabla_\theta \log \pi_\theta(da|x) e(x, a) \approx \frac{1}{m} \sum_{i=1}^{m} \nabla_\theta \log \pi_\theta(x, a_i) e(x, a_i). \tag{14}$$

Finally, we obtain a corresponding Robust Advantage Actor-Critic algorithms. We name it **W**asserstein **R**obust **A**dvantage **A**ctor-**C**ritic algorithm with order $p$, described in Algorithm 1 and Algorithm 2. Algorithm 1 is actually an inner loop that certifies the extent of perturbations, while Algorithm 2 finds the optimal policy in a normal way. Let the learning rates satisfy the Robbins-Monro condition (Robbins & Monro, 1951), and $\beta_1 = o(\beta_2)$, $\beta_2 = o(\beta_3)$, $\beta_3 = o(\beta_4)$, and via the multi-time-scales theory (Borkar, 2008), the convergence to a local minimum can be guaranteed.

## 5 EXPERIMENTS

In this section, we will verify WRAAC algorithm in Cart-Pole environment [2]. State space has four dimensions, including cart position, cart velocity, pole angle and pole velocity at tip. There are only two admissible actions: left or right. The target is to prevent the pole from falling over.

Our baseline includes the ordinary Advantage Actor-Critic algorithm. Policies are learnt under the default environment for WRAAC and the baseline. Then, we test the performances of these two policies under different environmental dynamics. We change the simulated environmental parameters such as gravity or pole-length to emulate different test dynamics. Note that the unit change on gravity and pole-length will result in different extents of the dynamic's robustness.

We apply WRAAC algorithm of order 2, and fix the degree of dynamical robustness at $\delta = 10$. For each quadruple $(x, a, r, y)$, if $y$ is not the last state of the trajectory, we set initial $\lambda$ be 0 and initial $z$ be $y + \delta \times (0, \frac{1}{\sqrt{26}}, 0, \frac{5}{\sqrt{26}})$ (designed according to the simulated dynamics of Cart-Pole). If $y$ is the last state, we set $\lambda \equiv 0$ and $z \equiv y$. The baseline policy and WRAAC are tested in environments with different gravity or different pole-length, shown in Figure 1 and Figure 2.

Remember that different parameters in the Cart-Pole environment have different effects to the dynamic's robustness. We can see that our robust algorithm changes smoothly as parameter changes,

---

[2]https://gym.openai.com/envs/CartPole-v0/

---

**Algorithm 1** Calculating Perturbations.

**Input:** $x \in \mathcal{X}$, $w$, $a \in A(x)$, $\delta \geq 0$, $\lambda \geq 0$, $e = 0$, $g_e = 0$, discount factor $\alpha$, order $p \geq 1$, $\kappa = 0$, learning rates $\beta_1$, $\beta_2$.
**for** $j = 1, 2, \cdots, n$ **do**
    collect roll-out $(x, a, c^j, y^j)$. $z^j \leftarrow y^j$.
    $z$ **update:**
    $g_z \leftarrow \nabla_z u_w(z) - \lambda(||z^j - y^j||^{p-2})(z^j - y^j)$,
    $z^j \leftarrow z^j + \beta_1 \cdot g_z$,
    $e \leftarrow e + c^j + \alpha[\lambda\delta + [u_w(z^j) - \lambda\frac{1}{p}||z^j - y^j||^p]] - u_w(x)$
    $g_e \leftarrow g_e + \alpha\nabla_w u_w(z) - \nabla_w u_w(x)$
    $\kappa \leftarrow \kappa + \frac{1}{p}||z - y^j||^p$,
**end for**
$\lambda$ **update:**
$g_\lambda \leftarrow \delta - \frac{1}{n}\kappa$,
$\lambda \leftarrow \lambda + \beta_2 \cdot g_\lambda$,
$e = \frac{1}{n}e$
$g_e = \frac{1}{n}g_e$
**Output:** $e$, $g_e$, $\lambda$.

---

**Algorithm 2** Wasserstein Robust Advantage Actor-Critic Algorithm with Order $p$.

**Input:** $x \in \mathcal{X}$, $\theta$, $w$, $\delta \geq 0$, discount factor $\gamma$, order $p \geq 1$, learning rates $\beta_3$, $\beta_4$
**for** each step **do**
    $E = 0$, $g_E = 0$.
    **for** $i = 1, 2, \cdots, m$ **do**
        sample $a_i \sim \pi_\theta(\cdot|x)$;
        use Algorithm 1 and obtain $e$, $g_e$.
        $e_i \leftarrow e$
        $E \leftarrow E + e$
        $g_E \leftarrow g_E + g_e$
    **end for**
    $w$ **update:**
    $w \leftarrow w - \beta_3 \cdot (\frac{1}{m}E) \cdot (\frac{1}{m}g_E)$
    $\theta$ **update:**
    $g_\theta = \frac{1}{m}\sum_{i=1}^m \nabla_\theta \log \pi_\theta(x, a_i)e_i$
    $\theta \leftarrow \theta - \beta_4 \cdot g_\theta$
    **state update:**
    choose $a \sim \pi_\theta(\cdot|x)$, and collect roll-out $(x, a, c, y)$ .
    $x \leftarrow y$
**end for**
**Output:** $\theta$, $w$.

---

while the baseline plunges. When the perturbation of parameter reaches some level (related with the fixed $\delta = 10$), our robust policy keeps the pole from falling over for a longer time, which indicates that our algorithm does learn some level of robustness, compared with baseline. If the perturbation of parameter is small, the baseline performs better, due to the fact that the perturbed environment is close to the default environment.

## 6 CONCLUSIONS

In this paper, we investigate the robust Reinforcement Learning with Wasserstein constraint. The derived theoretical framework can be reformulated into a tractable iterated-risk aware problem and the theoretical guarantee is then obtained by building connection between robustness to transition probabilities and robustness to states. Subsequently, we demonstrate the existence of optimal policies, provide a sensitivity analysis to reveal the effects of uncertainty set, and design a proper two-stage

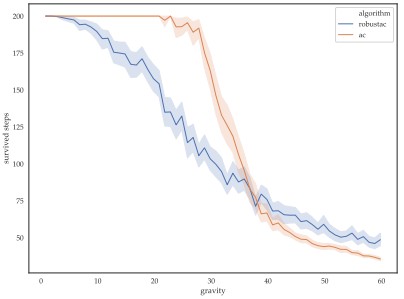 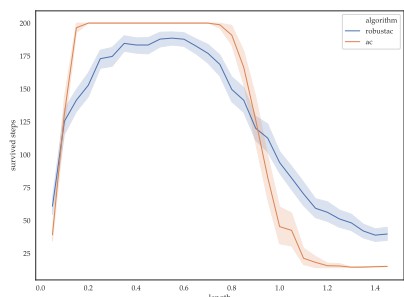

Figure 1: Robustness to gravity.      Figure 2: Robustness to length.

learning algorithm WRAAC. The experimental results on the Cart-Pole environment verified the effectiveness and robustness of our proposed approaches.

Future works may favor a complete study for the effects of the radius of Wasserstein ball in our WRAAC algorithm. We are also interested in studying robust policy improvement in a data-driven situation where we only have access to the set of collected trajectories.

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

## A  APPENDIX

The trajectory space $(\Omega, \mathcal{F})$, where $\mathcal{F}$ is the $\sigma$-algebra of $\Omega$, satisfies

- $\mathbb{P}_x^{\pi,g}(X_0 = x) = 1$,
- $\mathbb{P}_x^{\pi,g}(da|\omega_n) = \pi_n(da|\omega_n)$,
- $\mathbb{P}_x^{\pi,g}(dq|\tilde{\omega}_n) = \mathbb{1}(g_n(\tilde{\omega}_n) \in dq)$,
- $\mathbb{P}_x^{\pi,g}(X_{n+1} \in dx|\omega_n, a_n, q_n) = q_n(X_{n+1} \in dx|\omega_n, a_n)$.

Proof of Lemma 1:

*Proof.* (1) First, for $\{u_1, u_2\} \subset \mathbb{U}$, if $u_1 \geq u_2$, it's easy to have $Hu_1 \geq Hu_2$, i.e., the operator $H$ is monotone about $u$.
(2) For any real constant $C$ and $u \in \mathbb{U}$, we can verify that $H(u + C) = Hu + \gamma C$.
(3) For any $u_1 \in \mathbb{U}$, $u_2 \in \mathbb{U}$, there is $u_1 \leq u_2 + ||u_1 - u_2||_\infty$. Combining (1) and (2), we have $Hu_1 \leq Hu_2 + \gamma||u_1 - u_2||_\infty$, i.e., $Hu_1 - Hu_2 \leq \gamma||u_1 - u_2||_\infty$. Thus $||Hu_1 - Hu_2||_\infty \leq \gamma||u_1 - u_2||_\infty$. Furthermore, since $\gamma \in (0, 1)$, the operator $H$ has the contract property in $\mathbb{U}$ under $L_\infty$ norm.
(4) Via Banach fixed-point theorem, there exist an unique $u^* \in \mathbb{U}$ satisfying $Hu^* = u^*$. □

Proof of Theorem 1:

*Proof.* Due to Assumption 1, for any $u \in \mathbb{U}$, it is a measurable function on $\mathbb{K}_A$, an $(H^a u)(x)$ is lower semi-continuous w.r.t. $a$. Based on the measurable selection theorem (see Lemma 8.3.8 in (Hernández-Lerma & Lasserre, 2012b)), there is a deterministic Markov stationary policy $f \in \mathbb{F}$, satisfying $H^f u^* = Hu^* = u^*$. □

