# OpenReview forum: "Robust Reinforcement Learning with Wasserstein Constraint"
_ICLR.cc/2020/Conference — Reject_

### Official Review · AnonReviewer2 · 2019-10-22
**Official Blind Review #2**

**Rating:** 3

**Review:**

Summary:
In this paper, the authors study the problem of robust MDP (RMDP), where the feasibility set is defined as a Wasserstein ball around the reference transition probability. After formulating the problem, they first prove the contraction of the resulting operator and the existence of a deterministic stationary Markov optimal policy (Sec. 2.3). Then, they provide a sensitivity analysis of the optimal value function w.r.t. the radius and order of the Wasserstein ball (Sec. 2.4). Finally, they propose an actor-critic algorithm to solve the Wasserstein RMDP problem (Sec. 3) and evaluate its performance using simple experiments (Sec. 4).

Comments:
- Section 2.3, which the authors call it Main Result, is not that novel. The Wasserstein RMDP is just a state-action-rectangular RMDP with convex ambiguity set, and Lemma 1 and Theorem 1 are known to be true for such RMDPs.
- The sensitivity analysis section (Sec. 2.4) is interesting and useful, although it could have been written much better.
- The algorithm is new, although its section (Sec. 3) is not well-written. There is no analysis for the convergence of the algorithm either. It is a multi-time-scale stochastic approximation algorithm. Similar policy gradient and actor-critic algorithms have been derived in risk-sensitive MDPs for mean-variance, mean-VaR, and mean-CVaR optimization (see the references below), but such algorithms for RMDPs is relatively new.
- The experimental results are very simple and not very convincing.
- I would suggest that the authors put their emphasis on the algorithm and try to explain it much better. And support it better with more comprehensive and convincing experimental results. This would definitely improve the quality of the paper.


References on risk-sensitive MDPs:
1) A. Tamar, D. Di Castro, and S. Mannor. “Policy Gradients with Variance Related Risk Criteria”. ICML-2012.
2) Prashanth L.A. and M. Ghavamzadeh. “Actor-Critic Algorithms for Risk-Sensitive MDPs”. NIPS-2013.
3) Y. Chow and M. Ghavamzadeh. “Algorithms for CVaR Optimization in MDPs”. NIPS-2014.
4) A. Tamar, Y. Glassner, and S. Mannor. “Optimizing the CVaR via Sampling”. AAAI-2015.
5) A. Tamar, Y. Chow, M. Ghavamzadeh, and S. Mannor. “Policy Gradient for Coherent Risk Measures”. NIPS-2015.


**Experience Assessment:**

I have published one or two papers in this area.

**Review Assessment: Checking Correctness Of Derivations And Theory:**

I assessed the sensibility of the derivations and theory.

**Review Assessment: Checking Correctness Of Experiments:**

I assessed the sensibility of the experiments.

**Review Assessment: Thoroughness In Paper Reading:**

I read the paper at least twice and used my best judgement in assessing the paper.

---

> ### Author Response · Authors · 2019-11-15
> **Response to Review #2**
>
> Thank you for your detailed comments and suggestions.
> 1) We've checked several papers about state-action-rectangular RMDPs such as [1~3]. First, those papers deal with discrete state-action space, which avoid the measurability issues of the generalizaton to continuous state-action space. Second, their ambiguity sets are based on likelihood metrics such as KL divergence, which requires the absolute continuity between probability measures. Wasserstein distance has a whole different definition compared with likelihood metrics, and we do not find such papers that can natually extend Bellman results to Wasserstein-ambiguity set so far. It would be great if you could provide specific references.
> 2) The convergence of the algorithm to a local minimum can be guaranteed by the multi-time-scales theory [4]. We will analyse the effciency of our algorithm in the future.
> 3) We have corrected typos in the revised version. We will try to put more emphasis on the algorithm and conduct more convincing experiments as you suggested.
>
> [1] Nilim A, El Ghaoui L. Robust control of Markov decision processes with uncertain transition matrices[J]. Operations Research, 2005, 53(5): 780-798.
> [2] Iyengar G N. Robust dynamic programming[J]. Mathematics of Operations Research, 2005, 30(2): 257-280.
> [3] Wiesemann W, Kuhn D, Rustem B. Robust Markov decision processes[J]. Mathematics of Operations Research, 2013, 38(1): 153-183.
> [4] Borkar V S. Stochastic approximation: a dynamical systems viewpoint[M]. Springer, 2009.

---

### Official Review · AnonReviewer1 · 2019-10-24
**Official Blind Review #1**

**Rating:** 3

**Review:**

In this work the authors studied the robust reinforcement learning problem in which the constraint on model uncertainty is captured by the Wasserstein distance. Inspired by the analysis in the distribution-ally robust setting, they derived several sensitivity conditions to study how the radius of the wasserstein ball and the order of the wasserstein distance affect the conservativeness in model uncertainty.  Similar to the robust MDP work, they also show that this robust RL problem has a robust optimal Bellman operator, and the optimal value function can be computed using  robust policy iteration, which can be extended to model-free actor critic algorithm when state/action spaces are large/continuous.

This work extends the wasserstein uncertainty modeling  to the robust MDP setting. However, I found the contribution rather limited/unclear. First, the proposed robust Bellman optimality result is standard and can be found in many robust MDP work when the uncertainty set is state-action rectangular. The major novel part here is therefore the sensitivity analysis that is specifically tied to wasserstein distance, which has limited novelty. Second, the robust actor critic algorithm is based on a multi-time scale minimax gradient approach, which is also quite standard in this field, and besides asymptotic convergence it is unclear how efficient this algorithm is. Third, the experiment in this paper is very limited (only based on cartpole, and is only compared with the non-robust AC algorithm) and its illustration on the effectiveness of this algorithm is rather limited. More comparisons with other robust MDP methods would be useful to understand how the value of the proposed wasserstein robust RL formulation.

**Experience Assessment:**

I have published one or two papers in this area.

**Review Assessment: Checking Correctness Of Derivations And Theory:**

I assessed the sensibility of the derivations and theory.

**Review Assessment: Checking Correctness Of Experiments:**

I assessed the sensibility of the experiments.

**Review Assessment: Thoroughness In Paper Reading:**

I read the paper at least twice and used my best judgement in assessing the paper.

---

> ### Author Response · Authors · 2019-11-15
> **Response to Review #1**
>
> Thank you for your detailed comments and suggestions.
> 1) As far as we know, the existing robust MDP work with state-action-rectangular uncertainty set, such as [1~3], deal with discrete state-action space and ambiguity sets in the form of likelihood regions. First, those papers avoid the measurability issues of the generalizaton to continuous state-action space. Second, the metrics they used to describe ambiguity sets requires the absolute continuity between probability measures. Wasserstein distance has a whole different definition compared with likelihood metrics. So far we do not find such papers that can natually extend Bellman results to Wasserstein ambiguity set. It would be nice if you could provide specific references.
> 2) The convergence of the algorithm to a local minimum can be guaranteed by the multi-time-scales theory [4]. We will analyse the effciency of our algorithm in the future.
> 3) We will conduct more convincing experiments in the future as you suggested.
>
> [1] Nilim A, El Ghaoui L. Robust control of Markov decision processes with uncertain transition matrices[J]. Operations Research, 2005, 53(5): 780-798.
> [2] Iyengar G N. Robust dynamic programming[J]. Mathematics of Operations Research, 2005, 30(2): 257-280.
> [3] Wiesemann W, Kuhn D, Rustem B. Robust Markov decision processes[J]. Mathematics of Operations Research, 2013, 38(1): 153-183.
> [4] Borkar V S. Stochastic approximation: a dynamical systems viewpoint[M]. Springer, 2009.

---

### Official Review · AnonReviewer3 · 2019-10-24
**Official Blind Review #3**

**Rating:** 3

**Review:**


This work aims to produce reinforcement learning methods that are ‘distributionally robust’. They approach this by assuming the transition function may vary between elements of the domain distribution and extend some recent results (Blanchet & Murthy, 2019) to give some theoretical results (contraction and optimal deterministic policy) and an actor-critic algorithm. This is an important research area and the work takes a very nice approach. However, the clarity of the work and the empirical results could both use some work.

There are a great many grammatical errors throughout the paper. The nature of the errors suggests a non-native speaker, which I really do not want to discourage, but this needs a proofread and edit. One exception, there is an actual typo just after assumption 1 “dfnied”, but actually better grammatically to just remove the word entirely.

In the intro, the citation of Mannor et al. 2004; 2007 for what is essentially a description of the gap between simulation to real transfer is a very strange choice. Is it possible this was a mistake?

The description leading up to the main result could be made much clearer. I went through this section, the main results, and the proofs in the appendix. Things look good, although I do feel like (as just said) it could be made much better/clearer. One snag I hit was in the proof of Lemma 1, everything is good except I don’t know why “u_1 \le u_2 + \gamma \| u_1 - u_2 \|_\infty” must hold, but I might have simply missed something obvious so please help me out.

The experimental results are very minimal and not very convincing. It is not immediately clear to me that Figures 1&2 actually show that the robustac algorithm is more robust than ac. The performance is worse than ac, which is completely understandable, but for most of the environments ac is still significantly better.

Lastly, a small point to notice that this paper bleeds into the 9th page, and considering the amount of contributions (largely down to the two theoretical results and a very small experiment) I do not think the extra space is warranted. This could be compressed and actually come out stronger as a result of being forced to be clearer and more concise.

Update:

Thank you for your response, especially the fix to the proof. I will keep my score, but do believe that further experiments and small adjustments to the writing will see a future version of this accepted.


**Experience Assessment:**

I do not know much about this area.

**Review Assessment: Checking Correctness Of Derivations And Theory:**

I assessed the sensibility of the derivations and theory.

**Review Assessment: Checking Correctness Of Experiments:**

I assessed the sensibility of the experiments.

**Review Assessment: Thoroughness In Paper Reading:**

I read the paper at least twice and used my best judgement in assessing the paper.

---

> ### Author Response · Authors · 2019-11-15
> **Response to Review #3**
>
> Thank you for your detailed comments and suggestions.
> 1) We have corrected typos you mentioned, including that in the proof of Lemma 1, which is actually “u_1 \le u_2 + \| u_1 - u_2 \|_\infty”.
> 2) In the intro, we want to point out that the gap between simulation (training environment) to real (target environment) transfer makes the learned strategies sub-optimal, which is actually the motivation of Mannor et al. 2004; 2007. In their papers, the bias and variance resulted from the gap are learned to modify value function estimation and improve robustness of their results.
> 3) In figures 1&2, when the perturbation of parameter is large, our robust policy keeps the pole from falling over for a longer time, which indicates that our algorithm is more robust than AC.
> 4) In the future, we will conduct more convincing experiments and make our paper clearer and more concise as you suggested.

---

### Decision · Program_Chairs · 2019-12-19

**Decision:**

Reject

**Comment:**

This paper studies the robust reinforcement learning problem in which the constraint on model uncertainty is captured by the Wasserstein distance. The reviewers expressed concerns regarding novelty with respect to prior work, the presentation or the results, and unconvincing experiments. In its current form the paper is not ready for acceptance to ICLR-2020.